# Improved Prognostic Value in Predicting Long-Term Cardiovascular Events by a Combination of High-Sensitivity C-Reactive Protein and Brachial–Ankle Pulse Wave Velocity

**DOI:** 10.3390/jcm10153291

**Published:** 2021-07-26

**Authors:** Hack-Lyoung Kim, Woo-Hyun Lim, Jae-Bin Seo, Sang-Hyun Kim, Joo-Hee Zo, Myung-A Kim

**Affiliations:** Boramae Medical Center, Division of Cardiology, Department of Internal Medicine, Seoul National University College of Medicine, 5 Boramae-ro, Dongjak-gu, Seoul 07061, Korea; woosion@gmail.com (W.-H.L.); cetuximab@naver.com (J.-B.S.); shkimmd@snu.ac.kr (S.-H.K.); jooheezo@hanmail.net (J.-H.Z.); kma@snu.ac.kr (M.-A.K.)

**Keywords:** arterial stiffness, C-reactive protein, major adverse cardiovascular event, pulse wave velocity, risk stratification

## Abstract

Background: Both C-reactive protein (CRP) and arterial stiffness are associated with the development of cardiovascular disease (CVD). This study was performed to investigate whether a combination of these two measurements could improve cardiovascular risk stratification. Methods: A total of 6572 consecutive subjects (mean age, 60.8 ± 11.8 years; female, 44.2%) who underwent both high-sensitivity CRP (hs-CRP) and brachial–ankle pulse wave velocity (baPWV) measurement within 1 week were retrospectively analyzed. Major adverse cardiovascular events (MACE), including cardiovascular death, acute myocardial infarction, coronary revascularization, and stroke were assessed during the clinical follow-up. Results: During a mean follow-up period of 3.75 years (interquartile range, 1.78–5.31 years), there were 182 cases of MACE (2.8%). The elevated baPWV (≥1505 cm/s) (hazard ratio (HR), 4.21; 95% confidence interval (CI), 2.73–6.48; *p* < 0.001) and hs-CRP (≥3 mg/L) (HR, 1.57; 95% CI, 1.12–2.21; *p* < 0.001) levels were associated with MACE even after controlling for potential confounders. The combination of baPWV and hs-CRP further stratified the subjects’ risk (subjects with low baPWV and hs-CRP vs. subjects with high baPWV and hs-CRP; HR, 7.08; 95% CI, 3.76−13.30; *p* < 0.001). Adding baPWV information to clinical factors and hs-CRP had an incremental prognostic value (global Chi-square score, from 126 to 167, *p* < 0.001). Conclusions: The combination of hs-CRP and baPWV provided a better prediction of future CVD than either one by itself. Taking these two simple measurements simultaneously is clinically useful in cardiovascular risk stratification.

## 1. Introduction

Cardiovascular disease (CVD) is a leading cause of death and places a huge burden on our society worldwide [1]. Although various diagnostic and treatment methods have continuously been developed and applied to clinical practice, the prevalence of CVD is still high and the prognosis is poor. Identifying high-risk subjects who are more likely to develop CVD in the future and early implementation of active preventive strategies are critical to improving CVD prognosis [2]. While traditional risk factors represent cardiovascular risk well, they do not make all CVD incidences predictable [3]. In this respect, high-sensitivity C-reactive protein (hs-CRP), a sensitive marker for inflammation, has been recognized as a blood biomarker for predicting the occurrence of CVD. Inflammation plays a major role in the development and progression of atherosclerosis and the triggering of clinical CVD events [4]. Recent extensive evidence has suggested that high CRP levels are associated with higher risk of myocardial infarction, ischemic stroke, and sudden death [5,6,7,8,9].

Arteries stiffen with age and other risk factors, such as high blood pressure, hyperglycemia and smoking [10,11]. Of note, information on arterial stiffness is clinically valuable because it is associated with the occurrence of CVD, independent of traditional risk factors [12,13,14,15,16,17,18,19]. Of various methods of measuring arterial stiffness, pulse wave velocity (PWV) is most widely used in clinical and research fields [10].

Sometimes two test results are combined to improve the predictability of the prognosis [20,21]. Our group have also reported that PWV, along with noninvasive imaging studies, provides additional value in predicting the occurrence of CVD [22,23]. Although both hs-CRP and PWV have been used to predict CVD, there have been few studies on prognostic value by combining these two parameters. We hypothesized that the combination of hs-CRP and PWV would better predict the development of CVD. This study was performed to test this hypothesis.

## 2. Methods

### 2.1. Study Population

This single-center study was performed in a general hospital in a big city (Seoul, Korea). Between October 2008 and June 2018, a total of 8349 consecutive subjects who visited the cardiovascular center and underwent both brachial–ankle PWV (baPWV) and hs-CRP measurement within 1 week were retrospectively reviewed. The reasons for visiting the cardiovascular center vary widely, but it was not considered for study participation. The baPWV was measured by the attending physician as a routine part of a cardiovascular examination. Subjects with the following conditions were excluded from the study: (1) hs-CRP ≥ 10 mg/L (*n* = 1498) to rule out underlying active inflammatory conditions, (2) ankle-brachial index <0.9 or >1.4 (*n* = 85), (3) significant valvular dysfunction greater than mild degree (*n* = 43), (4) congenital heart disease (*n* = 6), (5) the presence of pericardial effusion (*n* = 12), and (6) atrial fibrillation and other uncontrolled arrhythmias (*n* = 133). Finally, 6572 subjects were analyzed in this study. This study was conducted in accordance with the declaration of Helsinki, revised in 2013. The study protocol was approved by the Institutional Review Board (IRB) of Boramae Medical Center (Seoul, Korea) and informed consent was waived due to the retrospective study design and the routine nature of information collected.

### 2.2. Data Collection

Body mass index was calculated as body weight in kilograms divided by the square of height in meters (kg/m^2^). Obesity was defined as body mass index ≥25 kg/m^2^. Hypertension was defined as previous diagnosis, current anti-hypertensive medications, or systolic and/or diastolic blood pressure ≥140/90 mmHg. Diabetes mellitus was defined as previous diagnosis, current anti-diabetic medications, or fasting blood glucose level ≥126 mg/dL. A person who smoked regularly in the last year was defined as a current smoker. Atherosclerotic cardiovascular disease (ASCVD) was defined as coronary artery disease including myocardial infarction and coronary revascularization, stroke, transient ischemic attack, and peripheral arterial disease [24]. After overnight fasting, blood samples were obtained in the antecubital vein and the blood levels of the following parameters were assessed: hemoglobin, creatinine, glucose, glycated hemoglobin, total cholesterol, low-density lipoprotein cholesterol, high-density lipoprotein cholesterol, triglyceride, and hs-CRP. Glomerular filtration rate was calculated by the Modification of Diet in Renal Disease (MDRD) study equation. Left ventricular ejection fraction was obtained by biplane Simpson’s method on transthoracic echocardiography. Information on cardiovascular medications was obtained, which included calcium channel blocker, beta-blocker, renin-angiotensin system blocker, diuretic, and statin.

### 2.3. baPWV Measurement

On the day of baPWV measurement, subjects were banned from smoking, alcohol, and caffeine-containing beverages such as coffee or green tea. Usual medications were not stopped and continued to be taken. The subjects rested in bed for about 5 min before the examination. The measurements were taken in a quiet closed room with constant temperature and humidity. The baPWV measured using a VP-100 analyzer (Colins, Komaki, Japan) [23,25]. After wrapping blood pressure cuffs around both upper arms and ankles, pressure waveforms of the brachial and tibial arteries were recorded with plethysmographic and oscillometric pressure sensors using occlusion/sensing cuffs. The time intervals between pressure waveforms of the brachial and tibial arteries (pulse transit time) were measured, and baPWV was automatically calculated at the estimated distance from the patient’s height [25]. The average of right and left baPWV measurements was used for analysis in this study. The baPWV was measured by an experienced operator. The coefficient of variation in baPWV measurement for intraobserver variability was 5.1% in our laboratory [26].

### 2.4. Clinical Events

The primary study endpoint, major adverse cardiovascular event (MACE), was composite clinical events consisting of cardiovascular death, non-fatal myocardial infarction, coronary revascularization, and stroke. Cardiovascular death included sudden cardiac death and death resulting from acute myocardial infarction, heart failure, stroke, cardiovascular procedures, cardiovascular hemorrhage, or other cardiovascular causes. Unexplained sudden death was considered cardiac death. Myocardial infarction was defined based on symptoms, electrocardiographic changes, elevation in cardiac troponin, and imaging results showing occlusive coronary artery lesions. Coronary revascularization included percutaneous coronary intervention and coronary bypass surgery. Stroke was diagnosed by neurologists using brain imaging study findings along with sudden neurological deficits.

### 2.5. Statistical Analysis

Continuous variables are expressed as mean ± standard deviation, and categorical variables are expressed as *n* (%). The means of continuous variables were compared using Student’s *t* test, and the prevalences of categorical variables were compared using Chi-square test between the 2 groups. Multivariable cox regression analyses were performed to find independent associations of hs-CRP, baPWV, and their combination with MACE. Variables with statistical significance in univariable analyses were used as independent variables in multivariable analysis. Receiver operating characteristic (ROC) curve analysis was used to obtain the cut-off value of baPWV predicting MACE. For the analysis of hs-CRP-related MACE, subjects were stratified into 2 groups by CRP levels: <3 mg/L vs. ≥3 mg/L [27,28]. Kaplan–Meier survival curve analysis was used to show event-free survival rates according to hs-CRP, baPWV, and their combination values. The log-rank test was used to test statistical significance. Additional prognostic value of hs-CRP and baPWV was assessed using global Chi-square scores. A *p* value of <0.05 was considered statistically significant. All statistical analyses were performed using SPSS 22.0 (IBM Corp., Armonk, NY, USA).

## 3. Results

### 3.1. Baseline Clinical Characteristics of the Study Subjects

During a mean follow-up period of 3.75 years (interquartile range, 1.78–5.31 years), there were 182 cases of MACE (2.8%), which included 14 cardiac deaths, 19 myocardial infarction cases, 118 coronary revascularization cases, and 49 stroke cases. The baseline clinical characteristics of the study subjects are shown in Table 1. Subjects with MACE were older and it was more common in males. Subjects with MACE had more cardiovascular risk factors, including hypertension, diabetes mellitus, cigarette smoking, and history of ASCVD compared to those without MACE. In laboratory findings, subjects with MACE had lower levels of hemoglobin, glomerular filtration rate, total cholesterol, and left ventricular ejection fraction, as well as higher levels of glucose and glycated hemoglobin than those without. Cardiovascular medications including beta-blocker, renin-angiotensin system blocker, and statin were more frequently prescribed in subjects with MACE than those without. Both baPWV (1833 ± 378 cm/s vs. 1588 ± 340 cm, *p* < 0.001) and hs-CRP (3.26 ± 2.75 mg/L vs. 1.97 ± 2.31 mg/L, *p* < 0.001) were significantly higher in subjects with MACE than those without (Figure 1).

### 3.2. Associations of baPWV, hs-CRP, and Their Combinations with MACE

ROC curve analysis showed that a baPWV of 1505 cm/s was the cut-off value predicting MACE, with a sensitivity of 83.5% and a specificity of 46.0% (Figure 2). Multivariable analyses showing associations of hs-CRP, baPWV, and their combinations with MACE are shown in Table 2. The elevated hs-CRP (≥3 mg/L) (hazard ratio (HR), 1.57; 95% confidence interval (CI), 1.12−2.21; *p* < 0.001) and baPWV (≥1505 cm/s) (HR, 4.21; 95% CI, 2.73−6.48; *p* < 0.001) levels were associated with MACE even after controlling for potential confounders. The combination of hs-CRP and baPWV further stratified the subjects’ risk (subjects with low hs-CRP and low baPWV vs. subjects with high hs-CRP and high baPWV; HR, 7.08; 95% CI, 3.76−13.30; *p* < 0.001). Kaplan–Meier survival curves demonstrated significant MACE differences according to hs-CRP (<3 mg/L vs. ≥3 mg/L, log-rank *p* < 0.001) and baPWV (<1505 cm/s vs. ≥1505 cm/s, log-rank *p* < 0.001) levels (Figure 3). In the Kaplan–Meier curve, by combining hs-CRP with baPWV, the subjects’ risk was further subdivided (Figure 4). Another combination of the hs-CRP level (≥3 mg/L) and clinical variables (age; sex; hypertension; diabetes mellitus; cigarette smoking; previous history of atherosclerotic cardiovascular disease; hemoglobin; glomerular filtration rate; total cholesterol; left ventricular ejection fraction; and the use of beta-blocker, renin-angiotensin system blocker, and statin) significantly increased prognostic value in predicting MACE (global Chi-square score, from 108 to 126, *p* < 0.001). Furthermore, the combination of baPWV (≥1505 cm/s) and hs-CRP + clinical variables had an incremental prognostic value in predicting MACE (global Chi-square score, from 126 to 167, *p* < 0.001) (Figure 5).

## 4. Discussion

Our results demonstrated that both increased hs-CRP and baPWV were independently associated with a higher risk for MACE in consecutive subjects visiting a cardiovascular center of a general hospital. More importantly, prognostic value was further improved by using a combination of hs-CRP and baPWV. Also, baPWV provided additional prognostic value in combination with clinical variables and hs-CRP in predicting MACE. To the best of our knowledge, this is the first study showing the improved prognostic value in predicting MACE by combining hs-CRP with baPWV.

### 4.1. Prognostic Value of CRP

CRP is a protein that increases synthesis in the liver against infection, inflammation, or tissue damage in our body. The level of CRP in the blood is proportional to the degree of synthesis in the liver [29]; thus, the blood level of CRP has been used for the diagnosis and treatment monitoring of infection or inflammatory disease. Due to the fact that the inflammatory response is deeply involved in the development and progression of atherosclerosis [4], CRP can also be a marker for atherosclerosis. In particular, inflammatory reactions in coronary plaques play an important role in plaque ruptures and subsequent acute atherothrombotic events [30]. There is also evidence that CRP is directly involved in the pathogenesis of atherothrombosis [31]. Many clinical studies have shown the association between elevated CRP and poor cardiovascular outcomes [5,6,7,8,9]. In line with these studies, our study also showed that baseline higher hs-CRP level (≥3 mg/L) was significantly associated with higher MACE incidences than those with lower hs-CRP level (<3 mg/L).

### 4.2. Prognostic Value of baPWV

Although carotid–femoral PWV (cfPWV) is the gold standard method for the non-invasive assessment of arterial stiffness [32], the clinical usefulness of baPWV, which is simpler to measure, is increasingly emerging [33,34]. The prognostic value of baPWV in predicting cardiovascular events has also been identified in many studies [13,17,18,19]. This study also showed that higher baPWV levels are independently associated with increased MACE incidences. The cut-off value of baPWV for predicting CVD has not yet been completely elucidated, and different cut-off values have been suggested in a different study population [18,35,36]. Therefore, we obtained the baPWV level that best predicts MACE in our study population through ROC curve analysis, and used it for survival analysis. It has been suggested that the cut-off value of baPWV that predicts future cardiovascular events depends on the subject’s cardiovascular risk. In subjects with relatively low cardiovascular risk, such as the general public, the cut-off value of baPWV is around 1500 cm/s, and in high-risk patients, such as those with coronary artery disease or diabetes mellitus, the cut-off value of baPWV is 1700–1800 cm/s [34]. Since the subjects of this study were consecutive subjects visiting the cardiovascular center, the overall cardiovascular risk was not high, and may be similar to or slightly higher than the general population, so the proposed cut-off in our study of 1505 cm/s is similar to the results of previous studies.

### 4.3. Combination of hs-CRP and baPWV

The main concern of our study was whether the ability to predict cardiovascular risk increases when hs-CRP and baPWV information are combined. Several studies performed by our group have shown incremental prognostic value of baPWV when combined with other non-invasive tests [23,26]. In the present study, we showed that the combination of hs-CRP and baPWV more accurately predicted MACE occurrence, and adding baPWV information to hs-CRP and clinical factors significantly increased prognostic power. CRP measurement is inexpensive and can be easily performed using venous blood. The measurement of baPWV is also non-invasive and simple, and it is useful especially for mass screening [33,34]. Given our findings and the simplicity of hs-CRP and baPWV measurements, the combination of the two measurements seems cost-effective to predict cardiovascular risks.

### 4.4. Study Limitations

Besides its retrospective study design, our study has several limitations. First, it is possible that a selection bias has occurred because we assessed subjects who performed both baPWV and hs-CRP measurement within 1 week. Second, many clinical variables that appear to be associated with the subjects’ cardiovascular risk were corrected in the multivariate analysis; however, we could not rule out the effects of possible uncorrected confounders. Lastly, our study population was restricted to Korean subjects, and generalization of our results to other ethnic groups is difficult.

## 5. Conclusions

Predicting cardiovascular risk is very important because early personalized treatment can improve a subject’s prognosis. Both hs-CRP and baPWV are known to be good predictors of cardiovascular events. In this study, it was shown that the combination of hs-CRP and baPWV provided better prediction of future CVD than either one by itself. Given that both hs-CRP and baPWV tests are easy and inexpensive to measure, taking the two simple measurements simultaneously is clinically useful for better cardiovascular risk stratification.

## Figures and Tables

**Figure 1 jcm-10-03291-f001:**
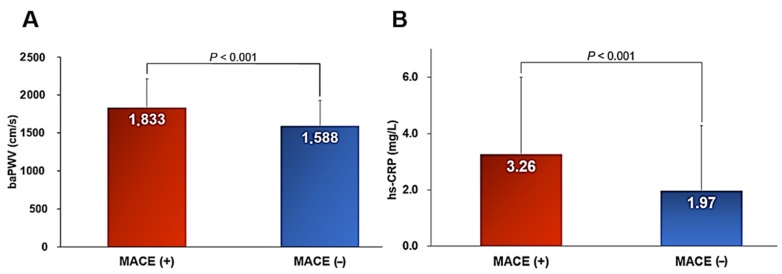
baPWV (**A**) and hs-CRP (**B**) values according to MACE. baPWV, brachial–ankle pulse wave velocity; hs-CRP, high-sensitivity C-reactive protein; MACE, major adverse cardiovascular events.

**Figure 2 jcm-10-03291-f002:**
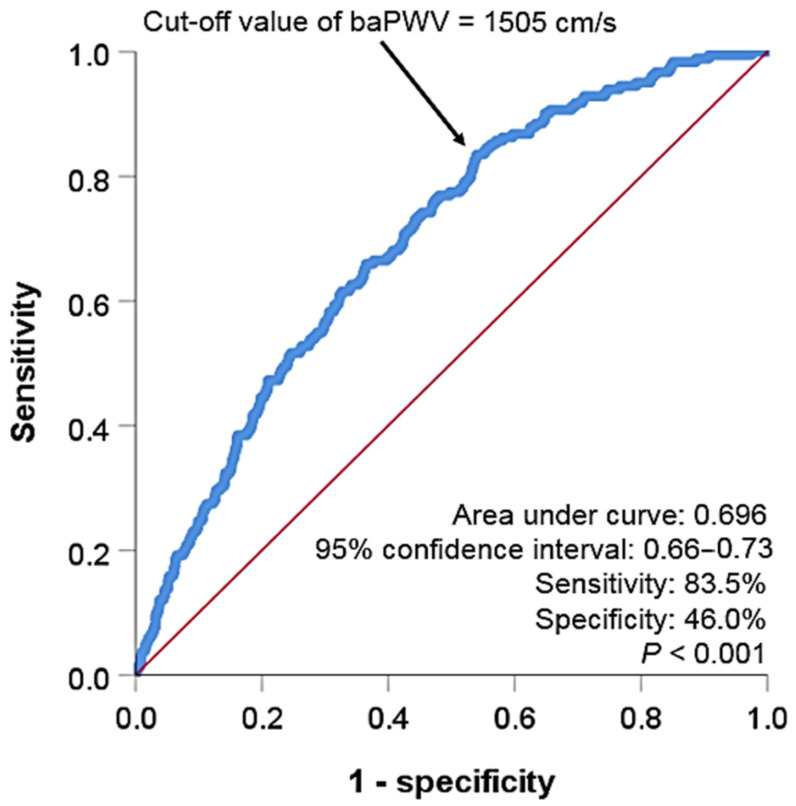
ROC curve analysis showing the cut-off value of baPWV for the MACE prediction. ROC, receiver operating characteristic; baPWV, brachial–ankle pulse wave velocity; MACE, major adverse cardiovascular events.

**Figure 3 jcm-10-03291-f003:**
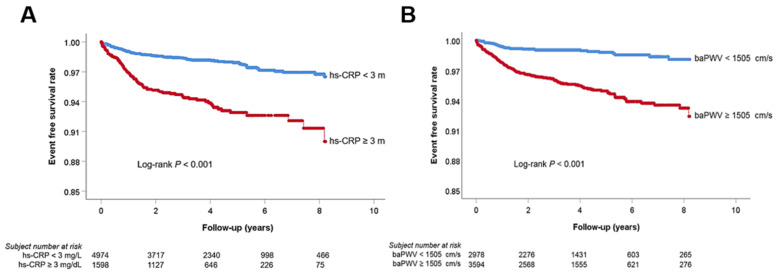
Kaplan–Meier survival curve analyses showing event-free survival rate according to hs-CRP (**A**) and baPWV (**B**). hs-CRP, high-sensitivity C-reactive protein; baPWV, brachial–ankle pulse wave velocity.

**Figure 4 jcm-10-03291-f004:**
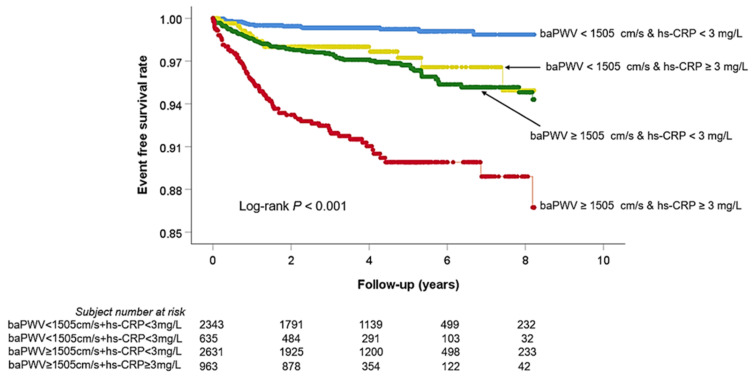
Kaplan–Meier survival curve analyses showing event free survival rate according to combination of hs-CRP and baPWV. hs-CRP, high-sensitivity C-reactive protein; baPWV, brachial-ankle pulse wave velocity.

**Figure 5 jcm-10-03291-f005:**
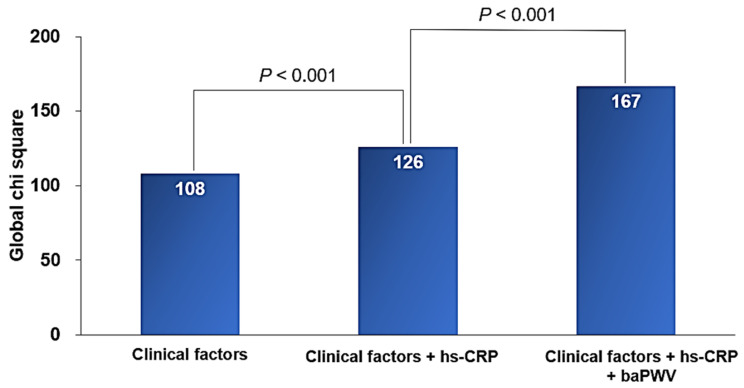
Incremental prognostic value of baPWV to clinical factors and hs-CRP. hs-CRP, high-sensitivity C-reactive protein; baPWV, brachial–ankle pulse wave velocity.

**Table 1 jcm-10-03291-t001:** Baseline clinical characteristics of study subjects.

Characteristic	Subjects with MACE (*n* = 182)	Subjects without MACE (*n* = 6390)	*p*
Age, years	64.7 ± 10.4	60.7 ± 11.8	<0.001
Male sex	123 (67.6)	3544 (55.5)	0.001
BMI, kg/m^2^	24.8 ± 2.9	24.9 ± 3.3	0.626
Cardiovascular risk factors			
Hypertension	105 (57.7)	3102 (48.5)	0.015
Diabetes mellitus	56 (30.8)	1471 (23.0)	0.015
Obesity (BMI ≥ 25 kg/m^2^)	79 (44.1)	2959 (46.4)	0.546
Cigarette smoking	49 (26.9)	1080 (16.9)	<0.001
Previous ASCVD	86 (43.7)	1470 (23.0)	<0.001
Laboratory findings			
Hemoglobin, g/dL	13.1 ± 2.0	13.6 ± 1.7	<0.001
GFR, mL/min/1.73m^2^	82.7 ± 26.1	87.0 ± 23.7	0.016
Fasting glucose, mg/dL	129 ± 52	119 ± 39	0.012
Glycated hemoglobin, %	6.61 ± 1.21	6.27 ± 1.07	0.018
Total cholesterol, mg/dL	159 ± 44	165 ± 38	0.039
LDL cholesterol, mg/dL	93.9 ± 41.6	96.4 ± 35.4	0.428
HDL cholesterol, mg/dL	47.7 ± 15.4	49.2 ± 12.8	0.212
Triglyceride, mg/dL	131 ± 82	131 ± 84	0.964
LV ejection fraction, %	60.4 ± 10.1	63.6 ± 9.1	<0.001
Cardiovascular medications			
Calcium channel blocker	22 (12.1)	1094 (17.1)	0.075
Beta-blocker	69 (37.9)	1438 (22.5)	<0.001
RAS blocker	82 (45.1)	1990 (31.1)	<0.001
Statin	121 (66.5)	2915 (45.6)	<0.001

MACE, major adverse cardiovascular event; BMI, body mass index; ASCVD, atherosclerotic cardiovascular disease; GFR, glomerular filtration rate; LDL, low-density lipoprotein; HDL, high-density lipoprotein; LV, left ventricular; RAS, renin-angiotensin system.

**Table 2 jcm-10-03291-t002:** Multivariable analyses showing the associations of baPWV, hs-CRP, and their combinations with MACE.

Variable	HR (95% CI)	*p*
hs-CRP ≥ 3 mg/L	1.57 (1.12−2.21)	<0.001
baPWV ≥ 1505 cm/s	4.21 (2.73−6.48)	<0.001
*hs-CRP + baPWV*		
hs-CRP < 3 mg/L and baPWV < 1505 cm/s	1	−
hs-CRP ≥ 3 mg/L and baPWV < 1505 cm/s	1.91 (0.89−4.10)	0.096
hs-CRP < 3 mg/L and baPWV ≥ 1505 cm/s	4.73 (2.57−8.70)	<0.001
hs-CRP ≥ 3 mg/L and baPWV ≥ 1505 cm/s	7.08 (3.76−13.30)	<0.001

Following clinical covariates were adjusted in each multivariable model: age; sex; hypertension; diabetes mellitus; cigarette smoking; previous history of atherosclerotic cardiovascular disease; hemoglobin, glomerular filtration rate; total cholesterol; left ventricular ejection fraction; and the use of beta-blocker, renin-angiotensin system blocker, and statin. baPWV, brachial–ankle pulse wave velocity; hs-CRP, high-sensitivity C-reactive protein; MACE, major adverse cardiovascular events; HR, hazard ratio; CI, confidence interval.

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
