# Peer review of "Improved Prognostic Value in Predicting Long-Term Cardiovascular Events by a Combination of High-Sensitivity C-Reactive Protein and Brachial–Ankle Pulse Wave Velocity"

_jcm, 2021, doi:10.3390/jcm10153291_

Round 1

Reviewer 1 Report

Kim and colleagues have demonstrated that the elevated brachial-ankle pulse wave velocity (baPWV) and C-reactive protein (CRP) levels were associated with major adverse cardiovascular events (MACE) even after adjustment for covariates using retrospective single center study data. In addition, they showed that the combination of baPWV and CRP further stratified the subjects’ risk. They concluded that the combination of CRP and baPWV provided better prediction of future cardiovascular death than either one by itself. The study concepts are easy to understand and there are some novelties in using combination of baPWV and CRP in this study.

Specific comments are followings.

Major Comments

#1. Methods: The explanation of study population in this study is insufficient. As with the discussion section, it is better to describe in method section that the study patients were an outpatient who visits the cardiovascular center. In addition, the authors should show the underling or comorbidities that affect PWV and CRP such as cancer and collagen disease.

#2. Methods: Heart rate is one of important factor to affect the baPWV. Please add heart rate to Table1 and multivariable cox regression analyses.  

Minor Comments

#1. Introduction: Please correct the format of reference 20, 21.

Author Response

Kim and colleagues have demonstrated that the elevated brachial-ankle pulse wave velocity (baPWV) and C-reactive protein (CRP) levels were associated with major adverse cardiovascular events (MACE) even after adjustment for covariates using retrospective single center study data. In addition, they showed that the combination of baPWV and CRP further stratified the subjects’ risk. They concluded that the combination of CRP and baPWV provided better prediction of future cardiovascular death than either one by itself. The study concepts are easy to understand and there are some novelties in using combination of baPWV and CRP in this study.

Specific comments are followings.

Major Comments

#1. Methods: The explanation of study population in this study is insufficient. As with the discussion section, it is better to describe in method section that the study patients were an outpatient who visits the cardiovascular center.

Response: We appreciate the reviewer’s valuable comments. Based on the comment, we added more detailed information on the nature of study population in the method section, as below.

Between October 2008 and June 2018, a total of 8,349 consecutive subjects who visited cardiovascular center and underwent both brachial-ankle PWV (baPWV) and hs-CRP measurement within 1 week were retrospectively reviewed. The reasons for visiting the cardiovascular center vary widely, but it was not considered for study participation.

In addition, the authors should show the underling or comorbidities that affect PWV and CRP such as cancer and collagen disease.

We could not show data on chronic inflammatory conditions in this retrospective cohort database. However, we excluded subjects with higher hs-CRP level ≥ 10 mg/dL. This might exclude most patients with active inflammatory conditions.

#2. Methods: Heart rate is one of important factor to affect the baPWV. Please add heart rate to Table1 and multivariable cox regression analyses.  

Response: We absolutely agree with the comment that heart rate affects baPWV value, but baseline heart rate was not influential to MACE in our study. In order to control confounding effects of clinical covariates with MACE, variables with statistical significance in univariable analyses were used as independent variables in multivariable analysis in our study. However, heart rates was not different between patients with and without MACE (68.6 ± 11.8 vs. 69.6 ± 12.2 bpm) in univariable comparison, and thus, we did not adjust for heart rate in multivariable analysis.

Minor Comments

#1. Introduction: Please correct the format of reference 20, 21.

Response: We appreciate the comment. We have corrected the format of the citation of the references appropriately.

Reviewer 2 Report

Kim et al has written an interesting article about the prognostic value of combining C-reactive protein and brachial-ankle pulse wave velocity. Both CRP and baPWV has been shown to have a potential as a prognostic marker, and the association between the two markers has also been established. This cohort study tries to investigate whether combining the two risk factors can increase the prognostic value.

The article is well written, and the results are clearly presented. I find the figures both informative and well displayed. However, I do have some questions to the methodology and design of the study.

Methods:

Study population:

How were the participants recruited? Were they referred to a cardiologist for a risk score, symptoms or were they recruited to this study only? Were the admitted due to symptoms? Please elaborate.

Data collection

I find the use of CRP to be inadequate described. What kind of CRP assay was used? Was it the same for the entire study period? If not, is it comparable? High sensitive assay? How was CRP > 3 defined as high elevated CRP, 99th percentile? Please elaborate.

CRP is known to increase after exercise. Is this taken into account? Do you know anything about activities the last 24 hours before the test? Was the test taken at the same time as the baPWV?  

Results:

The authors uses of a study specific cut off. This can be problematic. The cut-off in this study seems to be a little bit lower than the cut-off found in other studies. Please explain why the use of a study specific cut-off is better than earlier described cut-offs.

Author Response

Please see attachement.

Kim et al has written an interesting article about the prognostic value of combining C-reactive protein and brachial-ankle pulse wave velocity. Both CRP and baPWV has been shown to have a potential as a prognostic marker, and the association between the two markers has also been established. This cohort study tries to investigate whether combining the two risk factors can increase the prognostic value. The article is well written, and the results are clearly presented. I find the figures both informative and well displayed. However, I do have some questions to the methodology and design of the study.

Methods:

Study population:

How were the participants recruited? Were they referred to a cardiologist for a risk score, symptoms or were they recruited to this study only? Were the admitted due to symptoms? Please elaborate.

Response: This is a retrospective study. We enrolled consecutive subjects who visited cardiovascular center in a general hospital of big city. The reasons for visiting the cardiovascular center vary widely, but it was not considered for study participation. For more clarification, we revised and add some sentences about study subjects’ enrollment, as follows:

Between October 2008 and June 2018, a total of 8,349 consecutive subjects who visited cardiovascular center and underwent both brachial-ankle PWV (baPWV) and hs-CRP measurement within 1 week were retrospectively reviewed. The reasons for visiting the cardiovascular center vary widely, but it was not considered for study participation.

Data collection

I find the use of CRP to be inadequate described. What kind of CRP assay was used? Was it the same for the entire study period? If not, is it comparable? High sensitive assay? How was CRP > 3 defined as high elevated CRP, 99th percentile? Please elaborate.

Response: We appreciate your valuable comments. To be precise, CPR used in this study was high-sensitivity CRP (hs-CRP) in all study subjects. We changed CRP to hs-CRP throughout the entire manuscript. The definition of elevated CRP was based on existing research results. Based on the results of many previously published studies, Giollabhui et al. suggested clinical range of CRP: healthy CRP: <3 mg/L; subclinical elevated CRP: 3~9 mg/L; and elevated CRP: > 10 mg/L (Brain, Behavior, and Immunity 2020;87:898–900). Also, CRP > 3 mg/L was used as indicator of high cardiovascular risk in many clinical studies (Pearson et al. Circulation 2003;17:499-511; Ridker et al. Circulation 2003;107:391-397; Laaksonen et al. Eur Heart J 2005;26:1783-1789; Ong et al. Am J Epidemiol 2013;177:1430-1442). We additionally added some of these references in the method section.

CRP is known to increase after exercise. Is this taken into account? Do you know anything about activities the last 24 hours before the test? Was the test taken at the same time as the baPWV?  

Response: We agree with your comment that exercise or physical activity can have impact on both CRP and baPWV values. However, unfortunately, data on exercise or physical activity was not available in our retrospective cohort database.

Results:

The authors uses of a study specific cut off. This can be problematic. The cut-off in this study seems to be a little bit lower than the cut-off found in other studies. Please explain why the use of a study specific cut-off is better than earlier described cut-offs.

Response: We appreciate your comment about important issue. It has been suggested that the cut-off value of baPWV that predicts future cardiovascular events depends on the subject’s cardiovascular risk. In subjects with relatively low cardiovascular risk, such as the general public, the cut-off value of baPWV is around 1,500 cm/s, and in high-risk patients such as those with coronary artery disease or diabetes mellitus, the cut-off value of baPWV is 1,700~1,800 cm/s (Kim et al. Front Cardiovasc Med 2019;6:41). Since the subjects of this study were consecutive subjects visiting the cardiovascular center, the overall cardiovascular risk was not high, and may be similar to or slightly higher than the general population, so proposed cut-off in our study of 1,505 cm/s is similar to the results of previous studies. We think that this is an important issue that should be addressed in our study. Therefore, we added following description in the discussion section of revised manuscript, marked in yellow.

It has been suggested that the cut-off value of baPWV that predicts future cardiovascular events depends on the subject’s cardiovascular risk. In subjects with relatively low cardiovascular risk, such as the general public, the cut-off value of baPWV is around 1,500 cm/s, and in high-risk patients such as those with coronary artery disease or diabetes mellitus, the cut-off value of baPWV is 1,700~1,800 cm/s34). Since the subjects of this study were consecutive subjects visiting the cardiovascular center, the overall cardiovascular risk was not high, and may be similar to or slightly higher than the general population, so proposed cut-off in our study of 1,505 cm/s is similar to the results of previous studies.

Round 2

Reviewer 2 Report

Thank for your quick reply. The article is now ready to be published.